# #MaskOn! #MaskOff! Digital polarization of mask-wearing in the United States during COVID-19

**Jun Lang** [1], **Wesley W. Erickson** [2], **Zhuo Jing-Schmidt** [1]*

**1** Department of East Asian Languages and Literatures, University of Oregon, Eugene, Oregon, United States of America, **2** Department of Physics, University of Oregon, Eugene, Oregon, United States of America

* zjingsch@uoregon.edu

**Data Availability Statement:** The authors provided the search criteria used to query the API and retrieve the hashtag data in the Supporting information files. This will allow others to access the same data analyzed in this study. The data are

## Abstract

The coronavirus disease 2019 (COVID-19) has caused an unprecedented public health crisis worldwide. Its intense politicization constantly made headlines, especially regarding the use of face masks as a safety precaution. However, the extent to which public opinion is polarized on wearing masks has remained anecdotal and the verbal representation of this polarization has not been explored. This study examined the types, themes, temporal trends, and exchange patterns of hashtags about mask wearing posted from March 1 to August 1, 2020 by Twitter users based in the United States. On the one hand, we found a stark rhetorical polarization in terms of semantic antagonism between pro- and anti-mask hashtags, exponential frequency increases of both types of hashtags during the period under study, in parallel to growing COVID-19 case counts, state mask mandates, and media coverage. On the other hand, the results showed an asymmetric participatory polarization in terms of a predominance of pro-mask hashtags along with an "echo chamber" effect in the dominant pro-mask group, which ignored the subversive rhetoric of the anti-mask minority. Notwithstanding the limitations of the research, this study provides a nuanced account of the digital polarization of public opinion on mask wearing. It draws attention to political polarization both as a rhetorical phenomenon and as a participatory process.

## Introduction

The coronavirus disease 2019 (COVID-19) has caused an unprecedented public health crisis worldwide. The WHO declared it a global pandemic on March 11, 2020 [1]. In the United States, the total number of confirmed cases surpassed 20 million and the total number of recorded deaths from COVID-19 approached 350,000 on January 1, 2021 [2]. During the pandemic, egregious violations of public health guidelines by high-profile government leaders repeatedly made headlines, underscoring the politicization of the pandemic. At the center of this politicization was the use of face masks as a safety precaution, officially recommended by the Center for Disease Control and Prevention (CDC) on April 3, 2020 to mitigate the spread of the virus [3]. The White House, openly defying public health guidelines on mask wearing,

available for viewing on the OSF data repository site using the following link: https://osf.io/vgcrf/?view_only=1163e1cad81847f9a63115cae8da46e2 The data are also available from the corresponding author upon reasonable request and subject to Twitter's Terms and Conditions governing the sharing of Twitter data.

**Funding:** The author(s) received no specific funding for this work.

**Competing interests:** The authors have declared that no competing interests exist.

dominated news headlines at the first peak of the pandemic [4, 5]. Public eschewals of masks continued to generate news headlines, whether these occurred at Trump presidential campaign rallies or on the floor of the U.S. Senate [6].

While the media reports offered glimpses of the politicization of face masks during the pandemic, the extent to which public discourse was polarized on mask wearing has remained anecdotal, and the verbal representation of this polarization has not been explored. The impact of social media as information and advocacy platforms, especially during mass emergencies, is well known [7–9]. The role of social media as a public forum for political discourse has also garnered intense scholarly attention [10–16]. Twitter, in particular, has become "a freewheeling forum that allows instantaneous debate and commentary about virtually every subject under the sun" thanks to its hashtag feature, and plays a prominent role in the arena of public opinion where it has increasingly become a hotbed of political polarization [17–26].

A central theme that has emerged from the body of research on political polarization in social media is that users prefer interactions with like-minded others, reflecting social homophily, defined as "the principle that a contact between similar people occurs at a higher rate than among dissimilar people." [27]. Homophily generates a selective pattern of networked communication commonly known as the "echo chamber" effect by reinforcing preexisting views and limiting opposing views [17, 18, 23, 24, 28–31]. Furthermore, exchanges among value-aligned peers in echo chambers are much more frequent than cross-cutting exchanges. Colleoni et al. [28] identified 10 times as many Democrats as Republicans based on the content shared whereby Democrats exhibited higher levels of political homophily. Bastos et al. [29] found six times as many "in-bubble" interactions as "cross-bubble" interactions. Tsai et al. [30] found many more "echoers" ($n = 7754$) than "bridgers" ($n = 194$) in their sample. Similarly, Cossard et al. [31] found that the vaccine advocates outnumbered the vaccine skeptics by 54% and as the majority side of the debate they engaged in intragroup exchanges, entirely ignoring the postings of the vaccine skeptics. These findings indicate that value-aligned intragroup exchanges outnumber intergroup interactions and thereby consolidate majority views.

Research based on case studies of single issues has found echo chambers in discussions about various issues. These include political issues such as elections [32], German political parties [33], the impeachment of the former Brazilian President Dilma Rousseff [34], Brexit [29], and politically motivated boycotts [30], as well as issues that are not distinctly political, as in the Italian vaccination debate [31]. Studies on Twitter information sharing patterns across different issues have produced evidence that the echo chamber effect pertains primarily to political issues. Barberá et al. [35] found that retweets about political issues such as the 2012 election campaign, the 2013 government shutdown, and the 2014 State of the Union Address were shared within ideologically coherent echo chambers, but retweets about nonpolitical issues such as the 2013 Boston Marathon bombing, the 2014 Super Bowl, and the 2014 Winter Olympics were shared across partisan lines and displayed little ideological homophily. The study also found a dynamic development of polarization on the Newtown shooting characterized by a rapid change from cross-ideological exchanges to homophilic echo chambers.

A recent study released by the Knight Foundation (KF) [36] identified four ideological clusters of Twitter users: extreme left (10%), center left (57%), center right (8%), and extreme right (25%). The responses of these segments to six trending news issues, ranging from the Mueller investigation and white nationalism to North Korea and hurricanes, showed a stronger progressive dominance than the baseline ideology distributions, although the boundaries between political and nonpolitical issues are not as clear cut as those studied by Barberá et al. [35]. In her interpretation of the findings, Alexandra [37] maintained that users' baseline ideology distributions and responses to trending issues, along with their information diet and

referencing behavior, provide a perfect formula for polarization, stating "birds of a feather flock together, even in the digital skies of social media."

The KF study is unique in that it distinguishes the extreme segments from the center segments in addition to a left versus right grouping. This allows for a cross-cutting analysis of the relative weight of the center (both center left and center right) and the extreme (both extreme left and extreme right) as an index of political polarization at the level of participation in issues debates. We compared each of the six issues to the baseline in the KF data in terms of how much they engaged the ideological extreme versus the center and found evidence that the six trending issues are not equally polarizing. While responses to mass shootings, sexual harassment, and the Mueller investigation were consistent with the baseline, white nationalism, North Korea, and hurricanes elicited significantly more responses from the center segments and significantly less responses from the two extremes compared to the baseline (See S1 Table Ideology distributions across data sets based on Knight Foundation study [36]). Assuming that greater participation of the extreme in response to an issue indexes greater polarization of that issue, we found mass shootings, sexual harassment, and the Mueller investigation to be more polarizing than white nationalism, North Korea, and hurricanes, supporting the observation that political polarization at the level of participation is issue-specific. It is clear the binary distinction between what is political and nonpolitical is too simplistic to capture the gradience between the Mueller investigation with the strongest polarization (67% center vs. 33% extreme) and hurricanes with the weakest polarization (82.5% center vs. 18.5% extreme) in relation to the baseline. The political and nonpolitical dualism also fails to capture the contextual forces that shape public political discourse. For example, white nationalism is a distinctly political issue in American politics with heavy historical baggage [38]. In the KF data on Twitter discourse, however, it turned out to be one of the less polarizing issues in terms of participation, together with North Korea and hurricanes. In our view, this paradox highlights the need to analyze political polarization both as a rhetorical phenomenon and as a participatory process.

The COVID-19 pandemic knows no ideological divides, which makes it more like hurricanes than the Mueller investigation. It surpasses hurricanes in terms of the scope and depth of its public health and economic impacts and the far-reaching disruptions to daily life. As a distinctly nonpartisan crisis it could conceivably bring the whole country together in a collective response to the collective calamities. Yet the pandemic arrived in an election year, which inevitably set the stage for controversies. Mask wearing is a basic public health measure designed to protect all members of society by mitigating the spread of a dangerous virus. Notwithstanding the collective threats posed by the pandemic, there were a number of contextual forces that apparently deepened the partisan rift in public discourse about this policy. First, the mask guidelines touched upon a deep-seated tension between government authority and individual liberty, which is a perennial challenge in American political life [39]. Second, mask wearing was politicized in an election year marked by violent partisan vitriol, which was deliberately inflamed by populism and divisive rhetoric and amplified by the media. Third, mass uncertainties about the impacts of a first-in-a-lifetime pandemic were exacerbated by conflicting sources of information and misinformation [40, 41]. Fourth, the incumbent presidential candidate publicly downplayed the threats of the virus and repeatedly made news headlines for violating CDC mask guidelines, making mask-wearing even more controversial.

These contextual factors enable the hypothesis that discourse about mask wearing as a rhetorical phenomenon is deeply polarized. On the other hand, there is evidence that the majority of the general public supported masking with 88% saying masks should be worn in public at least some of the time [42]. There is also evidence that the Twitterverse is more progressive than the general public [36, 43]. Therefore, we hypothesize that equivalence between the two

sides of the rhetorical polarization will be unlikely. That is, polarization of mask wearing as a participatory process will be disproportionately divided between a pro-mask majority and an anti-mask minority. Based on previous observations of higher frequencies of intragroup exchanges in echo chambers than intergroup exchanges, we expect homophilic exchanges in the majority group made up of mask supporters. In this study, we test these hypotheses by examining mask-related stance-bearing Twitter hashtags posted by users based in the United States.

To this end, we collected a total of 412,959 stance-bearing English language hashtags about mask wearing posted between March 1 and August 1, 2020 by a total of 149,110 Twitter users based in the United States. Twitter hashtags are strings of characters preceded by the hash (#) character. They serve as popular topical markers that highlight the central ideas and themes of tweets [44]. As such they are handy tools for the linguistic expression and propagation of stance, defined as the "personal feelings, attitudes, value judgments, or assessments" of language users [45]. Hashtags are meme-like in the sense that they take on a life of their own through repeated sharing in largely unchanged forms and as such play a key role in viral trending of ideas. We examined the themes, frequencies, temporal trends and exchange patterns of the mask-related hashtags, how these trends related to coronavirus case counts, media coverage, and policy mandates, and compared the stance distributions in the hashtag data to the ideology distributions on Twitter. The results showed a sharply delineated rhetorical polarization of mask wearing characterized by emotionally charged semantic antagonism that escalated over the period of interest, along with the spread of the virus. On the other hand, we found that uses of pro-mask hashtags overpowered those of anti-mask hashtags above and beyond the level predicted by the baseline ideology distributions and the issue-based ideology distributions on Twitter whereby mask supporters stayed in "echo chambers" insulated from mask resistors.

## Materials and methods

### Data

The main data used in this study was aggregated population-level data of pro- and anti-mask hashtags collected from Twitter. Four sources of circumstantial data were analyzed in light of which to better understand the digital polarization of mask wearing: U.S. daily confirmed COVID-19 case counts, state executive mask mandates, search interest data from Google Trends, and news headlines of high-profile events related to COVID-19 management from EBSCO Newspaper Source. In addition, we drew on data from two existing studies: Twitter baseline ideology distributions and ideological segment proportions by issue from the Knight Foundation study [36], and Pew Research Center study on public opinion on mask wearing [42].

Twitter data was collected for all tweets posted between March 1 and August 1 that included any hashtags from an exploratory list of 15 mask-related hashtags identified in our initial observations of Twitter discourse about mask wearing in relation to COVID-19. This initial list included #WearAMask, #MaskUp, #WearADamnMask, #WearAFuckingMask, #WearYourMask, #WearYourDamnMask, #WearYourFuckingMask, #WearMasksSaveLives, #NoMasks, #MasksKill, #MaskOff, #MasksDontWork, #MasksAreForSheep, #SheepWearMasks, #MaskPropaganda, #Mask(s), #Masker(s), and #Antimasker(s). To avoid retrieval of duplicate items, we only included hashtags from original tweets and excluded retweets. We then refined our initial list by adding mask-related hashtags that were not in the initial list but were found in the retrieved tweets and were used more than 100 times and by removing low-frequency (<100) hashtags (e.g. #MaskPropaganda). We then excluded

hashtags that were bare nouns with no explicit indication of stance (e.g. #Mask(s), #Masker (s)). Using the refined list we conducted a further search and retrieved a total of 923,167 tokens of hashtags.

From this sample we selected hashtags with a U.S. origin based on self-reported locations, accepting common U.S.-associated names (e.g. USA, America), all state names including abbreviations, and U.S. location identifiers (e.g. NYC, Bay Area, San Fran, and Midwest). The locationally filtered data consisted of a total of 412,959 mask-related tokens of 35 distinct types of hashtags from a total of 149,110 users for analysis. Of the total users, 138,796 users tweeted exclusively pro-mask hashtags and 7,771 users tweeted exclusively anti-mask hashtags. There were 2,543 users who tweeted both types of hashtags. Of the total hashtags, 3,557 pro-mask hashtags were tweeted by 71 self-organizing advocates and providers of masks who took a pro-active approach to masking in response to COVID-19. The hashtags used by this group of users were retained in our data set for analysis. No handles in our data were on the published list of the IRA related bots [46]. All twitter data was collected using the Twitter API, custom Python scripts and packages, including GetOldTweets3, twitter_scraper, and requests_html, between July 27 and August 12, 2020. Our data was collected and used according to Twitter's Terms and Conditions. This study has been reviewed by the University of Oregon Research Compliance Services (RCS) under the 2018 Common Rule and determined to qualify for exemption under Title 45 CFR 46.104(d)(4).

The CDC COVID Data Tracker [47] and the Worldometer COVID-19 Data [2] were consulted for the number of daily new confirmed cases and cumulative case counts between March 1 to August 1, 2020.

The number and effective dates of state mask mandates were obtained by examining the official websites of state governments. Non-mandatory mask recommendations and mandates limited to business were excluded. By the beginning of August 2020, we counted 33 executive orders issued by governors and one executive order signed by the mayor of Washington D.C. (S2 Table Statewide mask mandates (DC. included) and S1 Fig Statewide mask mandates (33 states plus DC.)).

Google Trends indexes search interest on Google in a given time and region and provides a measure of issue salience [48–50]. Utilizing this online tool, we examined the results of keyword searches for "face mask" in the US between March 1 and August 1, 2020 in all search categories, including Web Search, News Search, YouTube Search, Image Search, and Google Shopping Search (S2 Fig Google Trends searches for "face mask").

We used EBSCO services to identify high-profile televised events related to mask wearing. This database provides full-text coverage of more than 40 US newspapers and full-text television and radio news transcripts from CBS News, CNN, FOX News, and NPR. Four groups of keywords ("mask" and "CDC", "mask" and "White House", "mask" and "Trump", as well as "mask" and "Pence") were used to search for news publications and transcripts between March and August, 2020, which yielded over 1000 news headlines. Non-U.S. media and headlines of low relevance were eliminated and a total of 62 headlines were obtained (S3 Table Mask-related news headlines).

## Methodology

This study employed both qualitative and quantitative analyses of the 35 stance-bearing mask-related hashtags to test the three hypotheses: 1) Twitter discourse on mask wearing is rhetorically polarized, 2) participation in the rhetorical polarization is asymmetric in favor of a pro-mask majority, and 3) an echo chamber effect is more likely in the pro-mask majority group than in the anti-mask minority group.

**Qualitative analysis.** Qualitative analysis focused on content analysis of the 35 mask-related hashtags. The two authors with a background in linguistics and two other linguists not involved in this study performed expert coding of the hashtags. They independently sorted the hashtags into two categories, pro-mask and anti-mask, with perfect intercoder agreement (Fleiss's *Kappa* = 1, $p < 0.001$, 95% CI (0.865, 1). The two coders involved in this study together conducted a granular semantic analysis of the hashtags in each category and grouped them into subcategories based on their lexical and grammatical resemblances, recognizing that subcategories thus identified have fuzzy boundaries while they differ in semantic focus [51]. Based on their semantic focuses, the pro-mask hashtags were sorted into four subcategories: 1) hortatives urging the use of masks or issuance of mask mandates (henceforth HORT), 2) assertions of the efficacy of masks (henceforth AS_efficacy), 3) assertions of the altruistic value of mask wearing (henceforth AS_altruism), 4) assertions of positive masculinity associated with mask wearing (AS_masculinity). The anti-mask hashtags were grouped into three subcategories: 1) rejection of masks and mask mandates (henceforth REJECTION), 2) insults to mask-wearers (henceforth INSULT), and 3) disinformation in the form of assertions of negative effects of mask wearing (henceforth DISINFORMATION). Here, disinformation is defined as any deceptive information with the potential for harm or intent to harm [13]. The other two coders not involved in the study were asked independently to sort the pro- and anti-mask hashtags into their respective subcategories identified by the authors, with the option of adding new categories if necessary. No new category was added by either coder. One coder produced incomplete sorting and the other produced complete sorting. The incomplete sorting was excluded from further consideration. Intercoder reliability between the two authors' sorting and the sorting by the coder with the complete sorting was computed. The intercoder reliability for the sortings of pro-mask hashtags was found to be Cohen's Kappa = 0.91 ($p < 0.001$), 95% CI (0.784, 0.998), indicating a substantial agreement between the two sortings. The intercoder reliability for the sortings of anti-mask hashtags was found to be Cohen's *Kappa* = 1 ($p < 0.001$), 95% CI (0.628, 1), indicating a substantial agreement between the two judgments. The only disagreement was on #MaskItOrCasket between a HORT coding and an AS_efficacy coding. An agreement was reached in favor of the latter coding after consultation. Two of the authors also annotated the news headlines and classified them into four content categories: 1) CDC mask guidelines, 2) Compliance with CDC guidelines, 3) Mixed messaging, and 4) Violation of CDC guidelines. The intercoder reliability was robust with Cohen's *Kappa* = .959 ($p < 0.001$), 95% CI (0.866, 0.994). The discrepancies were resolved through discussion between the coders and in consultation with the third author.

**Quantitative analysis.** For quantitative analysis, we used RStudio (version 1.2.5033) and Python (version 3.8.5) as data analysis tools to cleanse, inspect, analyze, and visualize our data. R packages used in this study included *lubridate*, *fpp2*, *zoo*, *tidyverse*, *scales*, *GGally*, *ggplot2*, and *RColorBrewer*. We first compared our sample of hashtag data with the sample of general Twitter data used in a Pew survey [43] showing that the majority (80%) of tweets come from a small minority (10%) of users. In our data, the top 10% most active users accounted for 52% of the hashtags and the bottom 90% users accounted for 48% of the hashtags. A Pearson's chi-square test of independence was computed on the association between data type (Pew sample of general Twitter data vs. mask-related hashtags in the present study) and user power grouping (top 10% vs. bottom 90%). The result showed that our Twitter hashtag data was significantly less skewed than the general Twitter data based on the Pew survey in terms of tweet volume ($\chi^2 = 17.47$, $p < 0.001$).

For each category of the hashtags, we calculated the type frequency (i.e. the number of distinct hashtags) and token frequency (i.e. the total occurrences of each distinct hashtag). To understand the distributions of the different functional subcategories of the hashtags, we

calculated the relative frequency of each subcategory by dividing its token frequency by type frequency. To understand how the hashtag uses changed over time, we plotted the temporal trends of pro-mask and anti-mask hashtags as two categories, as well as the temporal distributions of individual hashtags in each category using seven-day moving averages (March 4 to July 28). We first plotted the trajectories of the nine most popular pro-mask hashtags and all anti-mask hashtags and compared the trends. To illustrate the difference in volume between the two categories, we plotted the top three hashtags of each category in a stacked area graph. We then computed exponential growth rates of hashtag uses over selected intervals that appeared linear when viewed on a log scale. To explicitly compute the parameters $a$ and $\lambda$ for each best-fit exponential $y = ae^{\lambda x}$ over the $n$ days $x_i$ and cumulative hashtag counts $y_i$, we follow reference [52] to avoid weighting small $x_i$ too strongly; this is done by minimizing the quantity

$$\sum_{i}^{n} y_i \left( \ln (y_i) - \ln (a) - \lambda x_i \right)^2 \tag{1}$$

which leads to best-fit parameters given by

$$\ln (a) = \frac{\sum_{i}^{n}(x_i^2 y_i) \sum_{i}^{n}(y_i \ln (y_i)) - \sum_{i}^{n}(x_i y_i) \sum_{i}^{n}(x_i y_i \ln (y_i))}{\sum_{i}^{n} y_i \sum_{i}^{n}(x_i^2 y_i) - \left( \sum_{i}^{n} x_i y_i \right)^2} \tag{2}$$

$$\lambda = \frac{\sum_{i}^{n} y_i \sum_{i}^{n}(x_i y_i \ln (y_i)) - \sum_{i}^{n}(x_i y_i) \sum_{i}^{n}(y_i \ln (y_i))}{\sum_{i}^{n} y_i \sum_{i}^{n}(x_i^2 y_i) - \left( \sum_{i}^{n} x_i y_i \right)^2}. \tag{3}$$

To examine the interaction between pro-mask and anti-mask users, we identified tweets that were direct replies to a parent tweet whereby both the parent tweet and the reply tweet contained at least one stance bearing mask-related hashtag, and performed a Pearson's chi-square of independence on the association between stance (pro-mask vs. anti-mask) and direction of reply (reply to pro-mask vs. reply to anti-mask tweets).

Building on the results of the response bias, we zoomed in on the types of hashtags used by the two groups of anti-mask tweeters, one replying to the pro-mask tweeters and the other replying to fellow anti-mask tweeters. We cross-tabulated the hashtags used by the two groups, focusing on information type (disinformation vs. non-disinformation) and stance of recipient (anti-mask vs. pro-mask) as categorical variables and performed a Pearson's chi-square of independence on the association between the two. The DISINFORMATION category includes two hashtags (#MasksDontWork, #MasksKill). The non-disinformation category includes REJECTION and INSULT.

A key to understanding the hashtag trends is to see how they related to the COVID-19 disease trends. We computed and plotted the pairwise correlations between the time series of the daily counts of both types of hashtags and daily confirmed COVID-19 case counts. To further contextualize the temporal trends of the hashtags in the time span, we plotted the temporal trends of the hashtags together with COVID-19 confirmed case counts, state mask mandates, Google search trends for "face mask", and mask-related news headlines, using daily data based on seven-day moving averages for COVID-19 case counts and mask-related hashtags, and rescaled monthly data for the other categories for which no daily data was available.

Finally, to gauge the degree of digital polarization of public stances toward mask wearing as a participatory process, we compared our hashtag data to two sets of existing data: 1) the Knight Foundation data on baseline ideological segments and issue-based ideological segments averaged over six issues on Twitter [36] and 2) Pew Research Center data on pro- and anti-mask attitudes in the general public [42].

**Table 1. 26 types of pro-mask hashtags ranked in token frequency.**

| Rank | Pro-mask Hashtags | Token frequency | Percentage |
|---|---|---|---|
| 1 | #WearAMask | 150,787 | 39.02% |
| 2 | #WearADamnMask | 67,589 | 17.49% |
| 3 | #MaskUp | 51,558 | 13.34% |
| 4 | #Mask(s)(4)All | 29,355 | 7.60% |
| 5 | #WearMask(s)(2)SaveALife | 13,708 | 3.55% |
| 6 | #MasksSaveLives | 11,516 | 2.98% |
| 7 | #Mask(s)On | 10,389 | 2.69% |
| 8 | #WearYourMask | 10,269 | 2.66% |
| 9 | #MaskItOrCasket | 5,728 | 1.48% |
| 10 | #MasksNow | 4,714 | 1.22% |
| 11 | #WearTheDamnMask | 4,500 | 1.16% |
| 12 | #WearMasks | 3,740 | 0.97% |
| 13 | #MaskUpAmerica | 2,517 | 0.65% |
| 14 | #WearAMaskPlease | 2,514 | 0.65% |
| 15 | #WearTheMask | 2,382 | 0.62% |
| 16 | #CoverYourFace | 2,351 | 0.61% |
| 17 | #RealMenWearMasks | 2,164 | 0.56% |
| 18 | #MaskMandate | 1,889 | 0.49% |
| 19 | #WearAFuckingMask | 1,838 | 0.48% |
| 20 | #MandateMasks | 1,819 | 0.47% |
| 21 | #WearYourDamnMask | 1,532 | 0.40% |
| 22 | #MaskingForAFriend | 1,295 | 0.34% |
| 23 | #MasksWork | 783 | 0.20% |
| 24 | #WearingIsCaring | 610 | 0.16% |
| 25 | #CoverYourFreakinFace | 598 | 0.15% |
| 26 | #WearYourFuckingMask | 245 | 0.06% |

## Results

Of a total of 35 distinct hashtags in the data, 74% ($n = 26$) were pro-mask and 26% ($n = 9$) anti-mask. Of the 412,959 tokens of hashtags, 93.6% ($n = 386, 390$) were pro-mask and 6.4% ($n = 26, 569$) were anti-mask. It is clear that pro-mask hashtags outnumbered anti-mask hashtags in both type and token frequencies (Tables 1 and 2). A Fisher's Exact Test on the association between stance type (pro- vs. anti-mask) and frequency type (type vs. token) was

**Table 2. Nine types of anti-mask hashtags ranked in token frequency.**

| Rank | Anti-mask Hashtags | Token frequency | Percentage |
|---|---|---|---|
| 1 | #NoMask(s) | 15,890 | 59.81% |
| 2 | #Mask(s)Off | 4,112 | 15.48% |
| 3 | #MasksDontWork | 1,970 | 7.41% |
| 4 | #MasksOffAmerica | 1,437 | 5.41% |
| 5 | #NoMaskMandates | 1,363 | 5.13% |
| 6 | #NoMaskOnMe | 965 | 3.63% |
| 7 | #MasksAreForSheep | 400 | 1.51% |
| 8 | #SheepWearMasks | 345 | 1.30% |
| 9 | #MasksKill | 87 | 0.33% |

**Table 3. Functional subcategories of pro-mask hashtags ranked by relative frequency.**

| Rank | Category | Type freq. | Token freq. | Relative freq. |
|------|----------|-----------|-------------|----------------|
| 1 | neutral_HORT | 13 | 274,284 | 21,098.80 |
| 2 | expletive_HORT | 6 | 76,302 | 12,717.00 |
| 3 | AS_efficacy | 4 | 28,932 | 7,233.00 |
| 4 | AS_masculinity | 1 | 2,164 | 2,164.00 |
| 5 | AS_altruism | 2 | 1,905 | 952.5 |

statistically significant ($p < 0.001$), showing lower than expected token frequency in relation to type frequency for anti-mask hashtags and the opposite trends for pro-mask hashtags. Thematically, the semantic opposition between pro- and anti-mask hashtags is especially clear in the antonymic pairs, e.g. #MaskOn vs. #MaskOff, #MasksSaveLives vs. #MasksKill, #MasksWork vs. #MasksDontWork, and #MaskMandate vs. #NoMaskMandates.

Table 3 shows the type, token, and relative frequencies of the four subcategories of the pro-mask hashtags. The majority of pro-mask hashtags fall in the subcategory HORT in which six types contain expletive attributives (e.g. *damn, fucking, freaking*). These expletive attributives are known as emotive intensifiers and serve the purposes of intensifying the affective strength of language and demanding attention to what is being said [53]. They help create an emotionally charged atmosphere in the urgent calls to action. The HORT subcategory was broken down into expletive hortatives and non-expletive or neutral hortatives. A Fisher's Exact Test showed a non-significant lower than expected token frequency relative to type frequency for expletive_HORT and the opposite for neutral_HORT ($p = 0.277$). Table 4 shows the type, token, and relative frequencies of the three subcategories of anti-mask hashtags with REJECTION on the top, followed by DISINFORMATION.

Concerning the trends of the pro-mask hashtags (Fig 1), the first crest occurred in early April with a sharp spike of #Mask(s)(4)All, coinciding with the CDC recommendation of face masks. The second crest occurred in late May with #WearAMask leading the way. This drops to a deep trough in early June, followed by the third and most dramatic crest involving simultaneous surges of multiple hashtags around mid June that peaked in late June and early July with the top three being #WearAMask, #WearADamnMask and #MaskUp. These went down and up again in mid July to a fourth crest, which dipped abruptly at the end of July. The mask-related news headlines form clusters that roughly coincide with the growth peaks of the hashtags, dropping into a hiatus in June with only two headlines on CDC reminders of mask guidelines before the next cluster around early July. As can be seen in the color coding of the headlines, White House violations of the CDC guidelines were intensely reported in the earlier months, interspersed with headlines on mix-messaging by the White House. There was a scattering of both types of content even in July when finally Trump donned a mask.

Unlike the pro-mask hashtags, the anti-mask hashtags showed no marked peak in early April but exhibited similar temporal trajectories to those of the pro-mask hashtags for the remaining time period (Fig 2). We see a similar upswing starting May, which reached the first

**Table 4. Functional subcategories of anti-mask hashtags ranked by relative frequency.**

| Rank | Category | Type freq. | Token freq. | Relative freq. |
|------|----------|-----------|-------------|----------------|
| 1 | REJECTION | 5 | 23,767 | 4,753.40 |
| 2 | DISINFORMATION | 2 | 2,057 | 1,028.50 |
| 3 | INSULT | 2 | 745 | 372.5 |

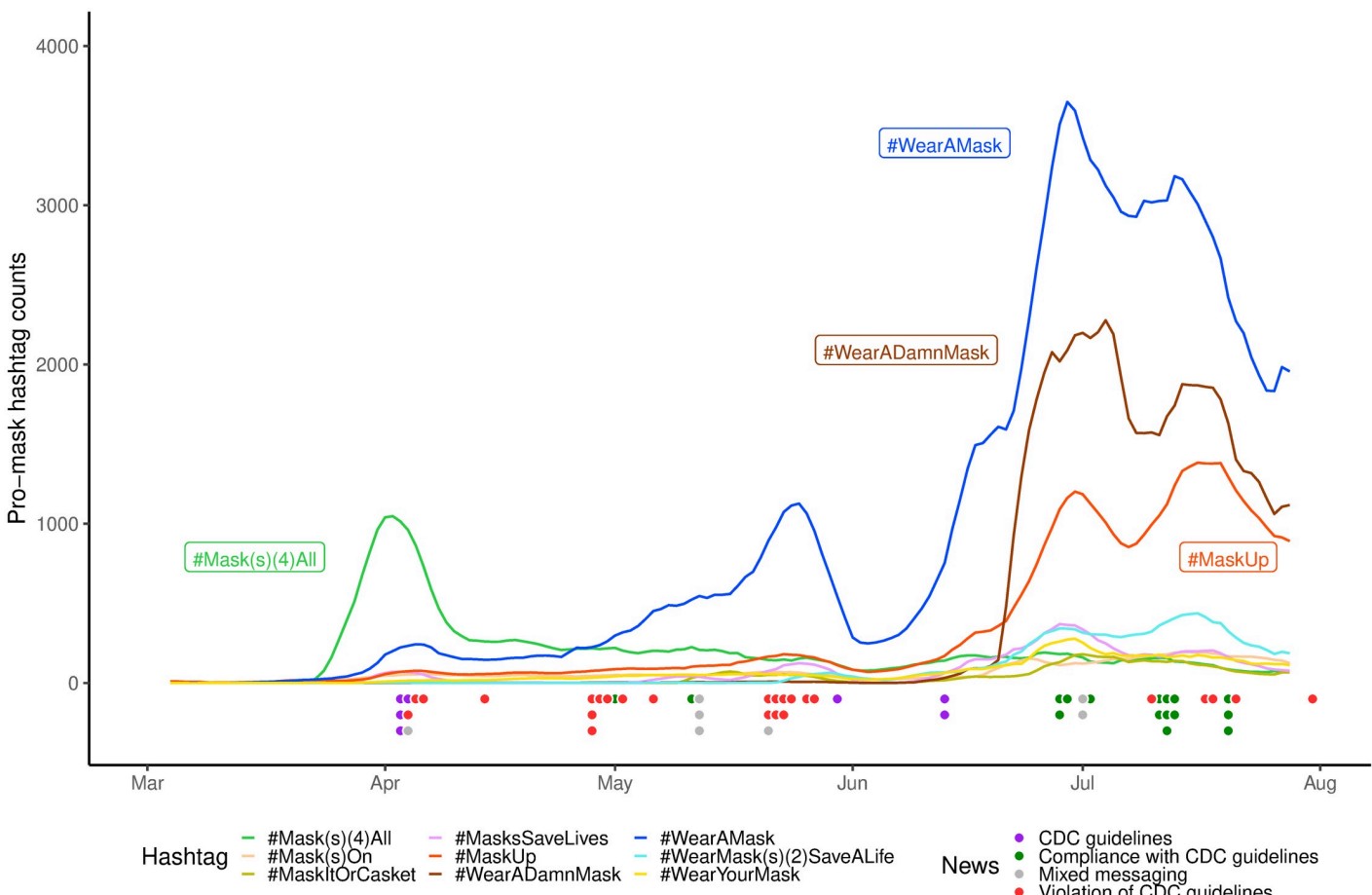

**Fig 1. Trends of pro-mask hashtag uses on Twitter.** This figure plots the 7-day moving averages of the daily rates of nine most popular pro-mask hashtags from early March to early August 2020. Along the timeline are shown news headlines in four categories: mask-related guidelines made by public health authority CDC, administrative actions compliant with the CDC guidelines, administrative violation of the CDC guidelines, and mixed messages from White House on masks.

peak in late May, falling in early June, and rising again more dramatically toward a second peak in late June and early July, followed by a dip before climbing to its highest peak in mid to late July. #NoMask(s) remained in the lead throughout the period, followed by #Mask(s)Off as a distant second. The same coincidence of news headline clusters with the peaks of the hashtags, as well as the news hiatus in June can be observed in Fig 2.

Fig 3 shows that these hashtags followed a very similar trajectory over time, which roughly tracks the temporal distributions of the news headline clusters. The large swaths of blue areas contrast with the thin stripes of red, showing the overwhelming dominance of mask supporters over mask resistors. It is clear that the volumes of the most popular pro-mask hashtags overshadow their anti-mask counterparts in extremely unbalanced proportions. Despite their visible difference in volume, both types of hashtags grew exponentially for the better part of the period under study. Fig 4 shows that the cumulative frequencies of both pro- and anti-mask hashtags experienced periods of exponential growth, with a period of faster exponential growth in March, followed by a period of slower exponential growth from April to early June.

The presence of two distinct phases of exponential hashtag growth shares some similarity to the observed exponential growth of COVID-19 cases in the U.S., which has a rapid exponential growth in February 2020 ($\lambda \sim 0.3$/day, $R^2 = 0.99$) calculated using Worldometer [2], before

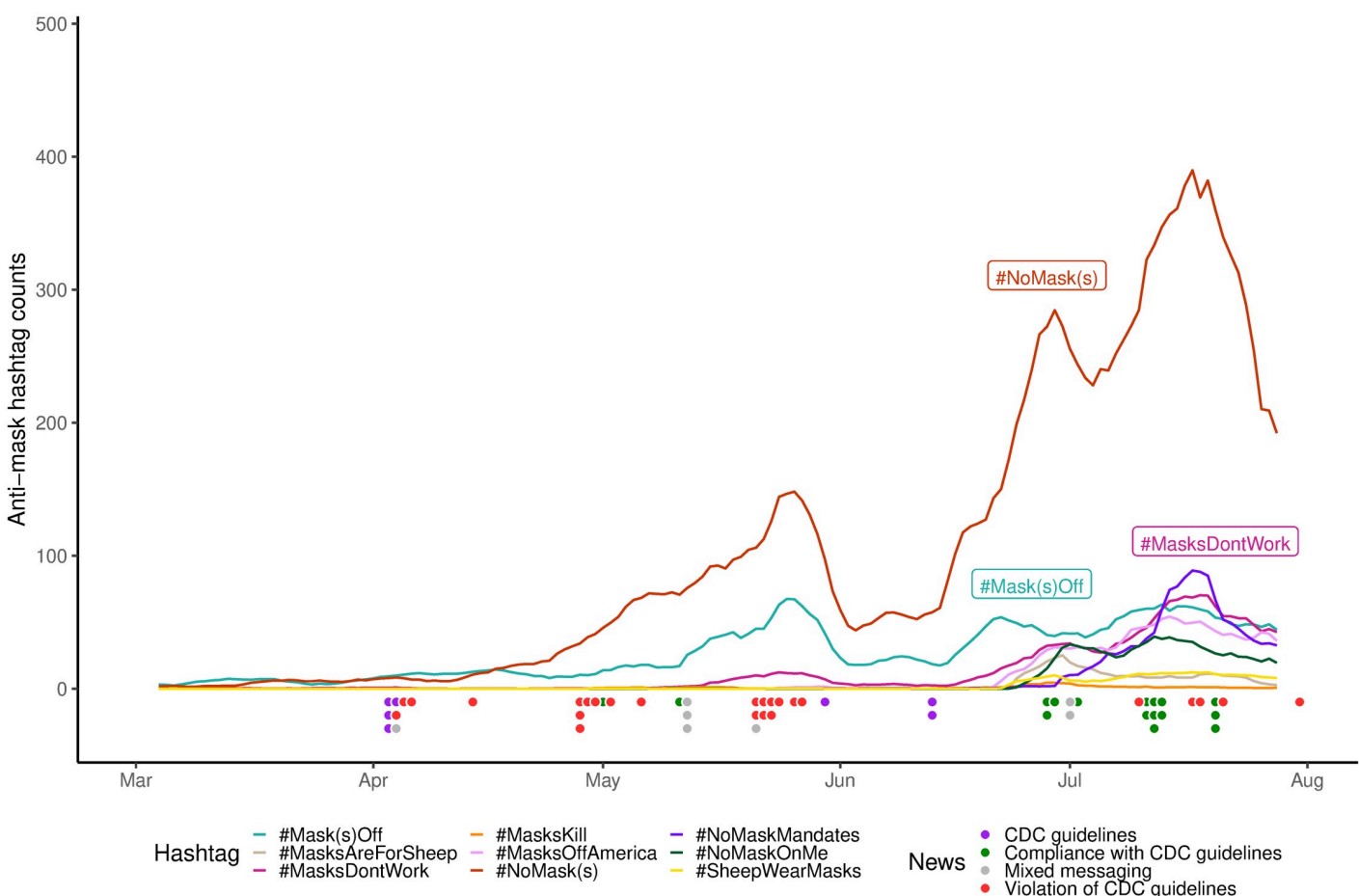

**Fig 2. Trends of anti-mask hashtag uses on Twitter.** This figure shows the 7-day moving averages of the daily rates of all anti-mask hashtags from early March to early August 2020. As Fig 1, along the timeline are shown media coverages on CDC recommendations and administrative actions related to masks.

dropping to a slower exponential growth ($\lambda \sim 0.01$/day, $R^2 = 0.96$) from April to October as lockdowns and precautions were implemented. It is also worth noting that the hashtag use of #H1N1 during the 2009 H1N1 pandemic also exhibited exponential growth ($\lambda \sim 0.01$, $R^2 = 0.99$), calculated using data from Chew and Eysenbach [54]. By contrast, tweet responses to short-term catastrophes do not appear to demonstrate long-term exponential growth, such as the 2010 Black Saturday fire in Australia [55], the 2015 Paris terrorist attacks [56], and the onset of a storm at the 2011 Pukkelpop Festival in Belgium [9]. It seems that the exponential growth of pandemics naturally leads to the exponential growth of related social media indicators, which is unsurprising given the pervasive threat of pandemics.

We found a total of 16,414 tweets that were direct replies to a parent tweet whereby both the parent tweet and the reply tweet contained at least one stance-bearing mask-related hashtag. Of these, 14,761 were pro-mask tweets replying to pro-mask tweets, 313 were pro-mask tweets replying to anti-mask tweets, 410 were anti-mask tweets replying to pro-mask tweets, and 930 were anti-mask tweets replying to anti-mask tweets. There was a significant association between tweet stance and the direction of response ($\chi^2 = 7,970$, $p < 0.001$). This biased direction of reply is further illustrated by examining user behavior (Fig 5). Defining pro-mask users as users that exclusively tweet pro-mask hashtags, and anti-mask users as users that exclusively tweet anti-mask hashtags, we found that many pro-mask users (teal points) only

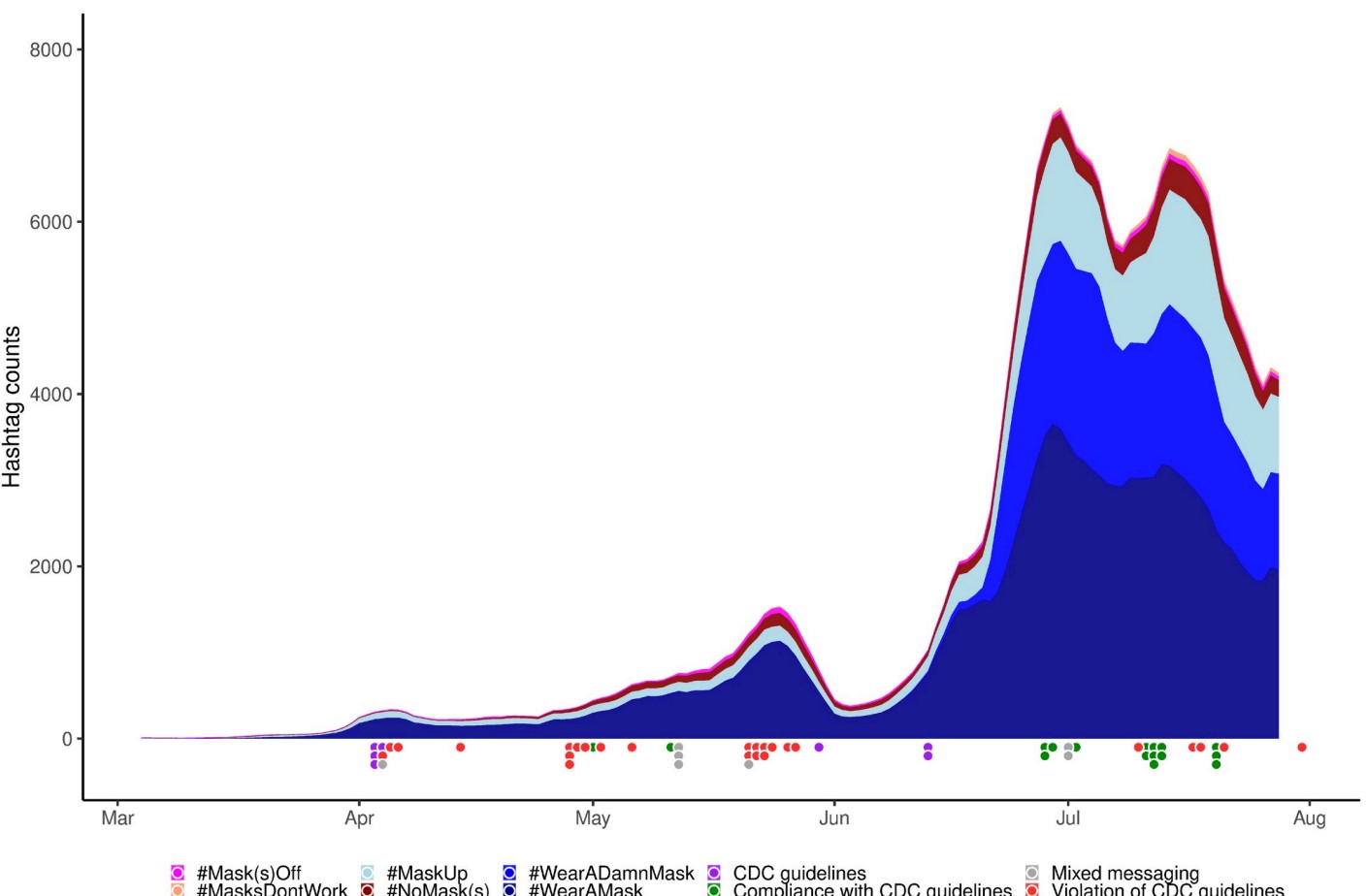

**Fig 3. Trends of top 3 anti- and pro-mask hashtags on Twitter.** This figure shows the 7-day moving averages of the three most popular anti-mask hashtags (#NoMask(s), #Mask(s)Off, #MasksDontWork) and pro-mask hashtags #WearAMask, #WearADamnMask, #MaskUp) in the period of interest in a stacked area chart.

replied to pro-mask tweets (Fig 5A). A smaller trend of anti-maskers (red points) only replying to anti-mask tweets is found along the vertical axis. The region near the origin (0, 0) contains many overlapping points, suggesting a concentration of low-volume replies to both pro- and anti-mask tweets. To investigate the clustering, we examined the normalized reply distributions for users of each opposing stance, which exhibit clear differences between pro-mask and anti-mask users: almost 50% of the anti-mask users replied at least once to pro-mask tweets (Fig 5B), whereas nearly 99% of the pro-mask users had 0 responses to anti-mask tweets (Fig 5C), indicating that anti-mask users are more likely to initiate intergroup communication while pro-mask users are more likely to engage in intragroup exchanges.

To further characterize the interactions between pro-mask and anti-mask Twitter users, we defined a tweet's stance $s$ as +1 if the tweet exclusively uses pro-mask hashtags and −1 if the tweet exclusively uses anti-mask hashtags. Then we defined a user's *response bias B* as the average stance of tweets that the user replied to using stance-bearing hashtags. That is

$$B = \frac{1}{N}\sum_{i}^{N} s_i,$$

(4)

where $s_i$ is the stance of each tweet the user replied to, and $N$ is the number of the user's

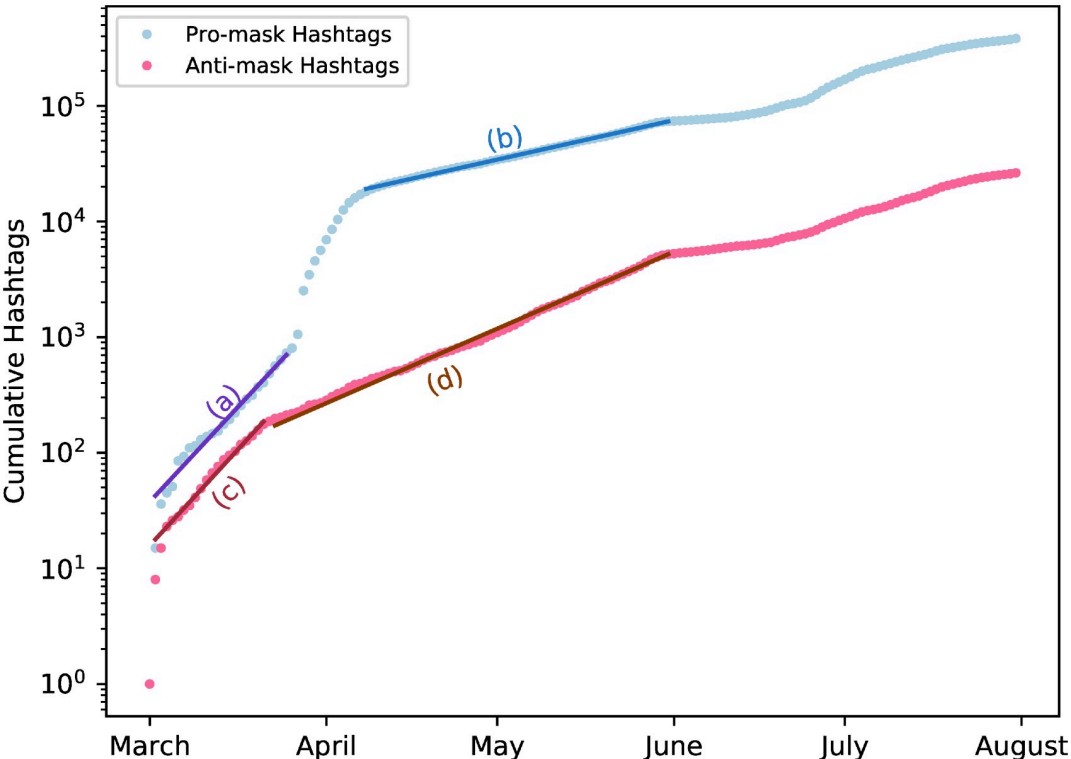

**Fig 4. Exponential trends of mask-related hashtags from March 1 to August 1 2020.** Fits (a-d) are best-fit exponentials with growth rate $\lambda$ over a given interval; for (a) $\lambda$ = 0.122/day from March 2 to March 26 with $R^2$ = 0.993, for (b) $\lambda$ = 0.025/day from April 8 to June 1 with $R^2$ = 0.998, for (c) $\lambda$ = 0.124/day from March 2 to March 22 with $R^2$ = 0.988, and for (d) $\lambda$ = 0.049/day from March 23 to June 1 with $R^2$ = 0.998.

stance-bearing replies to tweets containing stance-bearing hashtags. The response bias $B$ is bounded on the interval $[-1, +1]$ and characterizes the involvement in intra- and intergroup conversations of tweeters. $B = -1$ means the exclusive participation in replying to anti-mask tweets, whereas $B = +1$ indicates the exclusive participation in replying to pro-mask tweets. Values between $-1$ and $+1$ indicate the activity of conversing with both types of users. Fig 6 shows the probability distribution of the response bias for both pro- and anti-mask users, which indicates that pro-mask users almost exclusively interacted with pro-mask users, while anti-mask users are split into those that only interacted with anti-mask users, and those that only interacted with pro-mask users. For both pro- and anti-mask users, it is rare for a given user to engage with both pro- and anti-mask users, which demonstrates the polarized dynamic between tweeters.

Furthermore, the two groups of anti-mask tweeters that are split based on the stance of their recipients behaved differently in terms of the type of information conveyed by their hashtag choice. There was a statistically significant association between information type and stance of the recipient ($\chi^2$ = 4.74, $p$ = 0.029). Anti-mask tweeters who replied exclusively to pro-mask tweeters tweeted more tokens ($n$ = 83) of DISINFORMATION hashtags (e.g. #MasksDontWork and #MasksKill) in their replies than those who replied exclusively to other anti-mask tweeters ($n$ = 19).

To gauge the relationships between the time series of pro-mask hashtags, anti-mask hashtags, and daily confirmed COVID-19 case counts, we plotted each time series against the others. Fig 7 arranges these plots in a scatterplot matrix [57]. In the upper right half of the plot are

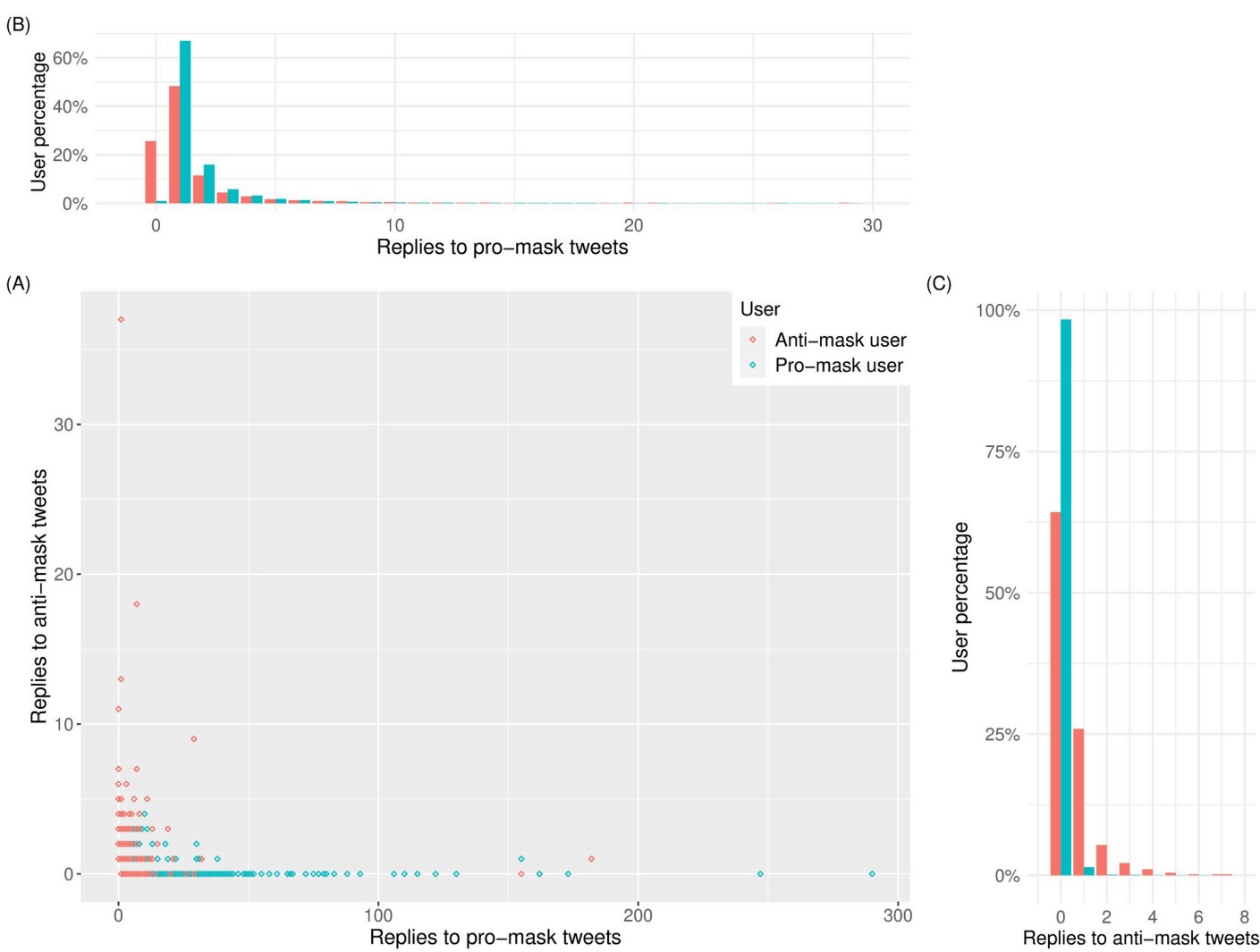

**Fig 5.** (A) Scatter plot of each user's replies to anti- and pro-mask tweets, with each red circle representing an anti-mask user and each teal circle representing a pro-mask user. The horizontal and vertical axes indicate the number of a user's replies to pro-mask tweets and to anti-mask tweets, respectively. (B) Distributions of replies to pro-mask tweets where the 0-reply teal bar is minimal in contrast to the 0-reply red bar representing over 20% of anti-mask users. (C) Distributions of replies to anti-mask tweets where the 0-reply teal bar representing 99% of pro-mask users sticks out on the horizontal axis, towering over the 0-reply red bar.

shown the correlations, while the scatterplots are shown in the lower left half. The first row, second column of Fig 7 shows a strong positive relationship between pro-mask and anti-mask hashtags ($r = .902$, $p < 0.001$). The first and second rows of the third column of Fig 7 shows that daily confirmed COVID-19 case counts were slightly more strongly correlated with anti-mask hashtags ($r = .836$, $p < 0.001$) than with pro-mask hashtags ($r = .822$, $p < 0.001$). In the density plot of pro-mask hashtags (first row, first column) and that of anti-mask hashtags (second row, second column), we found heavy (right) tailed distributions extending to large volumes, indicating occurrences of rare but significant events with high volume hashtag uses.

Finally, the temporal trends of COVID-19 case counts, state mask mandate counts, Google search interest in "face mask", and mask-related news headlines provide additional contextual cues for the temporal trends of the mask-related hashtags. Fig 8(A) shows that COVID-19 case counts plateaued from May to June, reaching a low in June when the hashtag counts took a

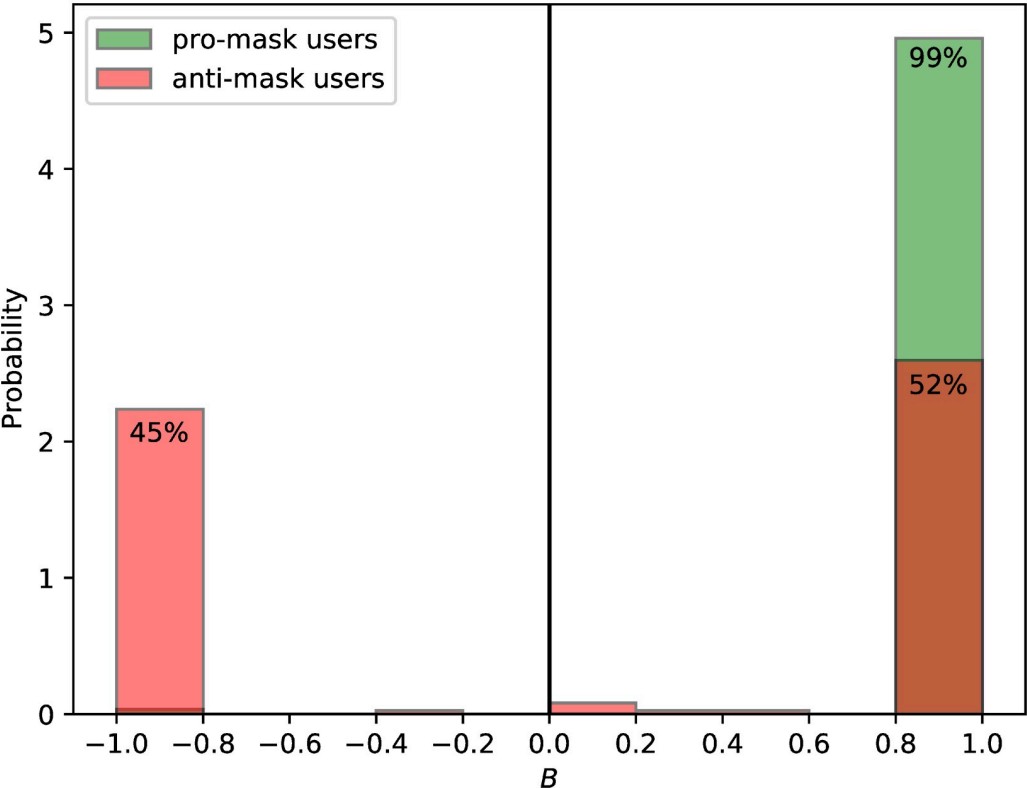

**Fig 6. The distribution of the response bias $B$ (Eq 4) for both pro-mask and anti-mask users.** The vast majority of users fall near the extremes of possible values of the response bias (i.e. within 0.2 of $B = +1$ or $B = -1$), with the percent of users in a given group indicated for each extreme bin.

bigger dip. Fig 8(B) shows that all the other circumstantial data categories took a much sharper plunge in June. After the June valley, all the data trends rose sharply in July.

To better understand the participatory polarization of mask wearing in Twitter hashtags, we compared the proportions of the opposing hashtag stances to the Twitter baseline ideology distributions reported in the KF study, assuming all segments except the extreme right were pro-mask. We found the pro-mask stance to be significantly overrepresented and the anti-mask stance to be significantly underrepresented in the hashtag data compared to the KF data ($\chi^2 = 13.781$, $df = 1$, $p < 0.0001$). When the hashtag data was compared to the KF issues subset data averaged over six trending issues, again assuming all segments but the extreme right were pro-mask, the association between dataset and stance proportion remained significant, though to a lesser degree ($\chi^2 = 4.31$, $df = 1$, $p = 0.038$). The proportion of pro-mask hashtags in our data exceeded expectations. It is clear public stances toward mask wearing in the hashtag data are strongly lopsided in favor of a pro-mask majority. The asymmetry is above and beyond the left skewing political landscape of Twitter and the average issue-based left-oriented stance, suggesting an overwhelming support for mask wearing in Twitter hashtags. By contrast, a comparison of our results with the Pew Research Center findings on public opinion on mask wearing [42] yielded no statistically significant difference ($\chi^2 = 2.198$, $df = 1$, $p = 0.138$), suggesting a consistency between Twitter hashtag data and the national survey of public opinion on mask wearing.

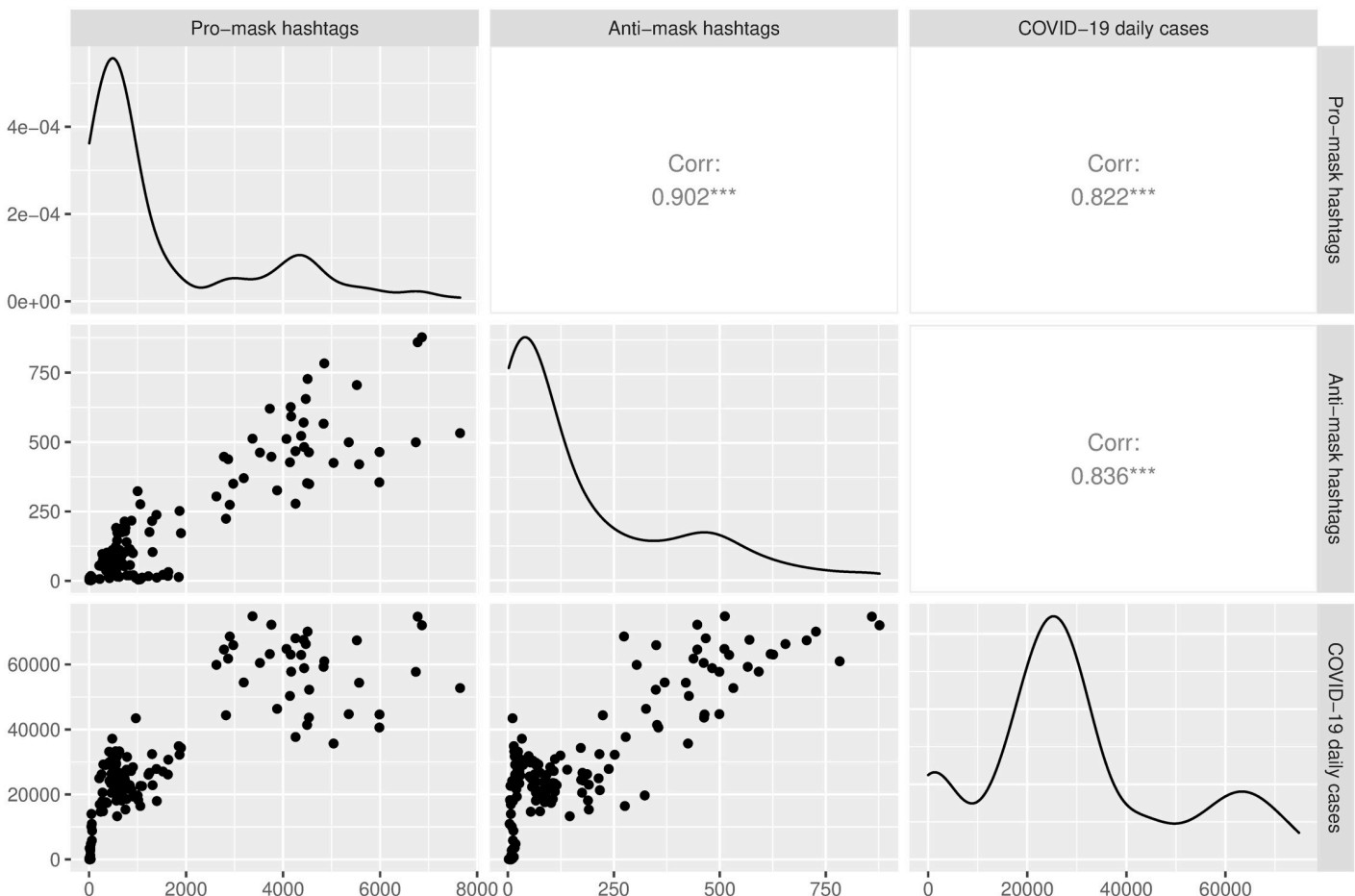

**Fig 7. Correlation matrix of the time series of pro-mask hashtags, anti-mask hashtags, and daily confirmed COVID-19 cases.** On the diagonal are shown the density plots, which visualize the distributions of the three variables over continuous intervals, with hashtag/case counts on the horizontal axis and number of days (normalized) using kernel probability estimation on the vertical axis.

## Discussion

Based on qualitative and quantitative analysis of aggregated Twitter hashtag data, our study yielded a complex picture of the digital polarization on mask wearing in the United States during the COVID-19 pandemic. On the one hand, we found a stark and persisting rhetorical polarization of public stance, characterized by an emotionally charged semantic antagonism between pro- and anti-mask hashtags. On the other hand, the sharp rhetorical polarization was accompanied by asymmetrical participation dominated by a pro-mask majority that was segregated in an "echo chamber" insulated from an anti-mask minority that attempted to infiltrate the pro-mask majority with disinformation. Taken together, our results demonstrate that the digital discourse on Twitter about mask wearing was rhetorically polarized whereby the rallying calls of the mask supporters were amplified by other mask supporters, and the battle cries of the mask resistors resonated with other mask resistors but were drowned out and ignored by a vocal and overwhelming pro-mask majority.

Both types of hashtags underwent exponential growth in the time span of interest (Fig 4). While there is evidence that pandemics tend to engender exponential increases of relevant social media activities, the parallel surges of the opposing hashtags suggest an escalation of the

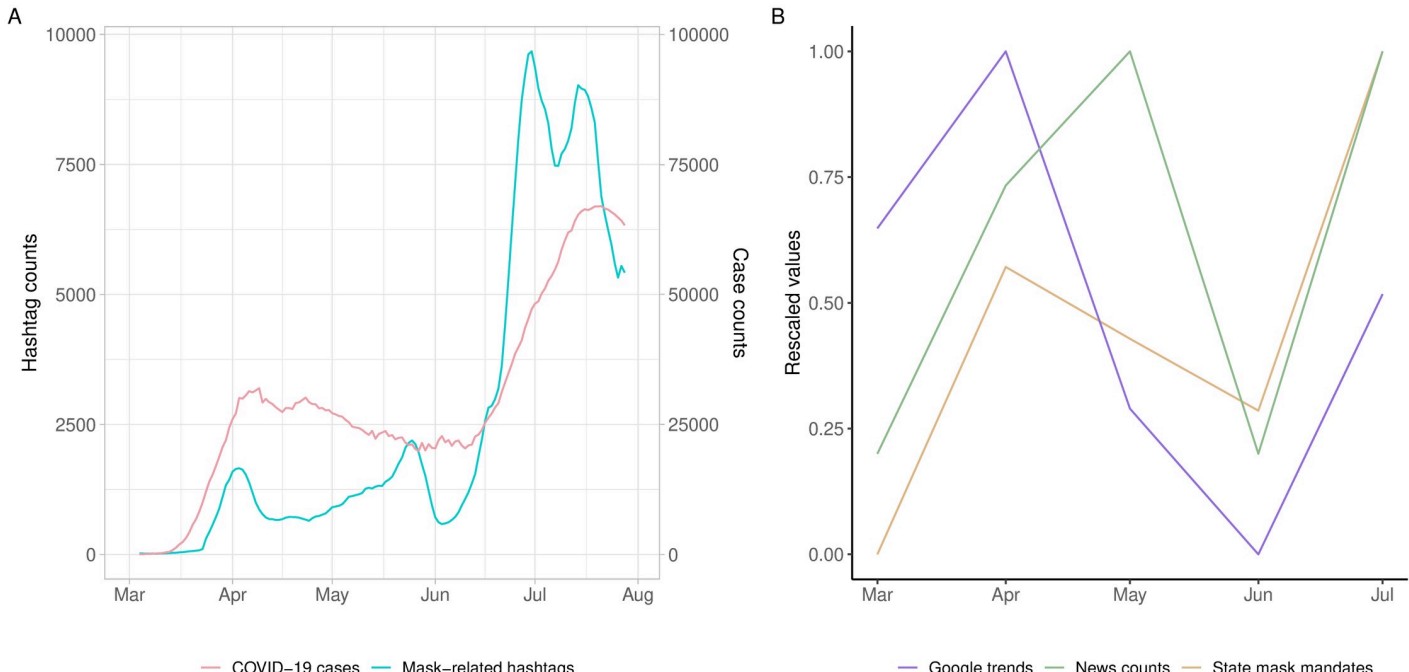

**Fig 8.** (A) Mask-related hashtag counts and daily confirmed COVID-19 case counts in the U.S. from early March to the end of July 2020. The figure plots the trends of mask-related hashtags on the left y-axis and the trajectory of daily confirmed COVID-19 cases on the right y-axis using seven-day moving averages for the two data series. (B) Three temporal trends from early March to the end of July 2020: (1) monthly statewide mask mandates, (2) monthly Google Trends searches for "face mask," and (3) monthly high-profile mask-related news headlines. The volume of each temporal trend is rescaled from 0 to 1 for the ease of comparison.

antagonistic rhetoric where neither side was willing to stand down. The strong correlations between the temporal trends of the hashtags and the temporal trends of COVID-19 case counts (Fig 7) suggest that uncertainties about the impacts of the pandemic grew in response to the spread of the virus. These uncertainties caused by the pandemic were undoubtedly exacerbated by an "infodemic" [41]. Faced with a first-in-a-lifetime pandemic, ordinary Americans scrambled to obtain information and found themselves grappling with conflicting sources of information [40]. The clusters of news headlines about leaders of the Trump administration violating mask guidelines, roughly concurrent with the early peaks of the hashtags, understandably magnified the rhetorical polarization of mask wearing, especially when a flurry of news reports on Trump flouting mask guidelines immediately followed the CDC issuance of those guidelines in early April. In response to the mismanagement of the pandemic, the pro-mask Twitterverse generated large volumes of tweets urging everyone to wear a mask whereas the anti-mask segment adhered to resistance modelled by the Trump White House. Apparently, the tensions between the two information sources aggravated the polarization of public opinion on mask wearing. It is worth noting that the CDC recommendation of facial masks to mitigate the spread of COVID-19 came after weeks of government health officials advising against the use of masks among the general public [58, 59]. This kind of contradiction in public health messaging contributed to the information crisis and may have deepened the public suspicion of the mask guidelines [60].

On the policy level, the surge of confirmed coronavirus cases in April compelled more state governors to institute mask mandates to slow the spread of the virus [61]. This can be seen in the April number of state mask mandates (Fig 8). On the other hand, mask mandates triggered intense push-backs in many U.S. states and localities, as widely reported in the media [62, 63].

It is reasonable to assume that the growing number of anti-mask hashtags on Twitter was in part the digital manifestation of those push-backs, exacerbated by the distrust of the news media by those who relied on Trump for information. The public health challenge to enforce mask wearing during a pandemic is not unique to the management of COVID-19. Nor is the politicization of facial masks a new phenomenon in American history [39]. The anti-mask battle cries on Twitter are the latest illustration of the perennial tensions between government imposed public health measures and individual autonomy, driven by an entrenched distrust of government authority that characterizes politics and civic culture in the United States [64–67]. Needless to say, populist attempts by leaders of the highest office to stoke public resentments and antiscience sentiments during the COVID-19 pandemic added fuel to the flame [68].

Our finding that the anti-mask stance persisted in a small portion (6.4%) of the hashtag data is consistent with the finding that a portion of Americans failed to follow public health recommendations without mask mandates and a small minority ($\sim$4%) continued to resist masks even when mandates were implemented [69]. Together, these findings support our hypothesis of a political polarization of mask wearing as a rhetorical phenomenon. Given accumulating evidence of the growing political polarization of the American electorate [42, 70–75], the polarization of mask wearing as a rhetorical phenomenon is clearly consistent with the partisan political climate in contemporary America.

The strong association between stance type and frequency type suggests that mask resistors were outgunned by mask supporters in the volume of hashtags they produced relative to the variety of hashtags at their disposal. The significant deviations of the hashtag data from Twitter baseline ideology distributions as well as from the issue-based ideology distributions further point to a strong skewness in terms of the participatory strengths of the pro- and anti-mask hashtags in the Twitter discourse about mask wearing. These findings support our hypothesis of an asymmetric participatory polarization of mask wearing in favor of a pro-mask majority. Together with the convergence between the hashtag data and the national survey data, the findings suggest that the American public stepped up in strong support of mask wearing despite the polarizing rhetoric about the policy and the news reports that magnified the rhetorical polarization. The predominance of pro-mask hashtag tokens seems to reflect the grave concern of the majority of the general public with the way the pandemic was managed in the U.S. An August 2020 Pew survey found that the majority of Americans (62% overall, 87% Democrats, 34% Republicans) say that the U.S. had trailed other developed countries in its response to COVID-19 [75]. The first peak of pro-mask hashtags coincided with the CDC official recommendation of face coverings on April 3. This finding is consistent with the results from national surveys that showed a positive effect of the CDC announcement of mask guidelines on public perceptions of the efficacy of face masks [76], on reported mask-wearing and buying behaviors [77], and on real-world observations of mask-wearing behavior among shoppers in retail locations [69].

The interactions between the tweeters, in particular the polarized distributions of their response bias are evidence of an "echo chamber" effect that reinforces the shared stance of like-minded users [18, 23, 24, 34, 35, 66, 78]. The low probability of a given tweeter engaging in cross-cutting interactions with both pro- and anti-mask users is a further indicator that tweeters with opposing views on mask wearing are insulated from one another. Our results show it was mostly pro-mask tweeters that stayed in an echo chamber, confirming the hypothesis that stance-congruent homophily is found in the dominant group. On this point, our finding is consistent with previous studies (e.g. [29–31, 36]). Because the larger group is more naturally an "echo chamber", fringe groups are much more likely to encounter a "mainstream" view than the reverse. It has been shown experimentally that exposure to opposing views on social media may create backfire effects that intensify political polarization [17]. Under the

backfire effects, the anti-mask users would be prone to increase their commitment to resistance to mask wearing. However, because they were severely outnumbered, their defiance and disinformation would have negligible impact on the pro-mask users as the dominant group. The lopsided participatory polarization as seen in our study and in other studies reveals a false balance or false equivalence often associated with opposing stances when the fringe group is given attention disproportionate to its size. This may be attributable to the general asymmetry in the automatic processing of affective information, known as the negativity bias whereby people pay more attention to negative events than positive and neutral events [79–81]. The role played by the media in creating this false balance by magnifying the anti-mask rhetoric should not be overlooked, given the persisting media bias toward negative news [82].

The directionality of disinformation on the efficacy of face masks (e.g. #MasksDontWork and #MasksKill) suggests the strategic use of semantic content for social manipulation, as shown in a recent study on bots targeting specific individuals during the Catalan Referendum of October 1, 2017 [83]. In our data, the overall volume of disinformation hashtags used in the unidirectional interactions targeting mask supporters is small and we have no evidence that they were generated by bots. To the extent that disinformation peddled by a small band of mask resistors failed to penetrate the "echo chamber" of mainstream support for masking, the peril of echo chambers might be "overstated", to use the word of Dubois and Blank [84]. This said, the greater media attention given to incendiary rhetoric including disinformation is often a forgotten part of the equation.

The monthly circumstantial data trends converged in June in a visible turning point (Fig 8B). The number of state mask mandates dwindled, Google search interests in face masks dropped, and the number of mask-related news headlines plunged. The converging downward trends suggest that state governments, news media, and the public all turned their attention away from the pandemic. This is when states started reopening or loosening restrictions in preparation of reopening. According to the June 2020 Pew survey, 40% of Americans say "the worst is behind us", a 14% jump compared to April [42]. The June turning point in the data captures a moment when the collective guard was down, corroborating the reported misperception that the nation was out of the woods. Around the same time, the murder of George Floyd by a Minneapolis police officer set off protests and unrest in major cities across the country, further shifting attention allocations in the public sphere [85].

Our study has several limitations. First, we did not examine the social network structure of the hashtag users and therefore were ill-prepared to make direct observations of the behaviors of the distinct groups based on information on their followers and mutual followers. Second, our analysis focused on hashtags as the central vehicle of viral ideas and did not include other contents in tweets, which were relevant for understanding public stance toward mask wearing and would have provided a fuller picture of the digital discourse on mask wearing. A sentiment analysis of these other contents would have greatly complemented our results. Third, we focused on reply patterns among the hashtags and did not examine other forms of information exchange on Twitter (e.g. retweets and mentions), which were relevant for gauging interactions between pro-mask and anti-mask tweeters. While providing a window on the relative interaction frequencies of pro- and anti-mask tweeters with robust data, this narrow focus certainly could not capture all interactions or lack of interactions between the tweeters, and may have limited the informativeness of our results, especially in the analysis of the echo chamber effect. Lastly, previous studies found that COVID-19 mask rhetoric in social media is gendered [86], that there is a gender difference in the likelihood of mask-wearing in public [69, 76], and that male leaders who value a macho image tend to eschew masks because they perceive mask wearing as emasculating [87]. One of the pro-mask hashtags in our data, e.g. #RealMenWear-Masks, was clearly designed to refute the view that mask-wearing diminishes masculinity. We

did not examine the gender identification of the tweeters in this study, and the extent to which the stances of the hashtags in our data are divided along gender lines awaits future investigation.

## Conclusion

Despite the demonstrated effectiveness of masks in mitigating the spread of the coronavirus disease 2019 (COVID-19), mask wearing as a safety precaution remained a source of controversy in the United States during the COVID-19 pandemic. Drawing on aggregated data from Twitter, this study provides evidence of a stark digital rhetorical polarization of public opinion on masking that was exceedingly unbalanced in terms of the participation of the opposing sides. Notwithstanding the various limitations, our study highlights the importance of looking at political polarization as an issue-specific problem that is at once a rhetorical phenomenon and a participatory process. That the predominance of the pro-mask stance exceeded the level of support for masking that could be predicted from the ideological partitions of Twitter users cautions against the false equivalence in the perception of the two sides of a political polarization. It suggests that a rhetorical polarization, no matter how uncivil, is not necessarily matched by a participatory process equally divided along partisan lines. This said, considering the exceptional nature of the COVID-19 pandemic in terms of the scope and severity of its impacts, the results found in this study offer us little solace when we contemplate the human cost of this crisis, which is yet to be measured. Nor do the findings offer a source of relief from the pervasive, digitally synchronized, rhetorical polarization in social media that erodes trust in institutions in an unprecedented way [13, 88]. This study underscores the importance of interdisciplinary research on public health discourse during a pandemic, and has implications for research on political polarization in social media.

## Supporting information

**S1 File. Search criteria used to query the API.**
(TXT)

**S1 Fig. Statewide mask mandates.** The figure shows the effective dates of executive mask mandates (33 states plus D.C.) on the x-axis and the cumulative number of mandates on the y-axis over the time period under study.
(PDF)

**S2 Fig. Google Trends searches for "face mask".** The figure shows the trajectories of weekly search interests for face mask on Google Trends in the U.S. over the time period under study. The vertical axis represents the relative interest to total search volume. Searches for face mask spiked in early April in all five search categories, coinciding with the CDC mask recommendation. After the sharp decline, search interests rebounded to a second peak in late June and increased further in mid-July in most categories except YouTube search.
(PDF)

**S1 Table. Ideology distributions across data sets based on Knight Foundation study [36].**
(PDF)

**S2 Table. Statewide mask mandates (DC included).**
(PDF)

**S3 Table. Mask-related news headlines.**
(PDF)

## Acknowledgments

We thank Vsevolod Kapatsinski for his comments on an earlier draft of this paper. We thank Thomas E. Payne and Eric Pederson for helping with the classification of the hashtags. All errors in the writing belong to the authors.

## Author Contributions

**Conceptualization:** Zhuo Jing-Schmidt.

**Data curation:** Jun Lang, Wesley W. Erickson.

**Formal analysis:** Jun Lang, Wesley W. Erickson.

**Methodology:** Jun Lang, Wesley W. Erickson, Zhuo Jing-Schmidt.

**Supervision:** Zhuo Jing-Schmidt.

**Visualization:** Jun Lang, Wesley W. Erickson.

**Writing – original draft:** Jun Lang, Wesley W. Erickson, Zhuo Jing-Schmidt.

**Writing – review & editing:** Jun Lang, Wesley W. Erickson, Zhuo Jing-Schmidt.

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
