## [Decision Letter · Decision Letter 0]

1 Mar 2021

PONE-D-21-03538

#MaskOn! #MaskOff! Digital polarization of mask-wearing in the United States during COVID-19

PLOS ONE

Dear Dr. Jing-Schmidt,

Thank you for submitting your manuscript to PLOS ONE. After careful consideration, we feel that it has merit but does not fully meet PLOS ONE’s publication criteria as it currently stands. Therefore, we invite you to submit a revised version of the manuscript that addresses the points raised during the review process.

In the revised version of the paper please address the reviewers' comments listed below. Additionally, please provide a more in depth discussion regarding the results, while better stating the limitations of the study.

We look forward to receiving your revised manuscript.

Kind regards,

Liviu-Adrian Cotfas

Academic Editor

PLOS ONE

Journal Requirements:

2. PLOS ONE has specific requirements for studies using personal data from third-party sources, including social media, blogs, other internet sources, and phone companies (https://journals.plos.org/plosone/s/submission-guidelines#loc-personal-data-from-third-party-sources). These requirements include confirming data are collected and used in accordance with the company or website’s Terms and Conditions, obtaining appropriate ethics or data protection body review, and ensuring appropriate consent from individuals whose data are used in research. In this case, please ensure that your Ethics statement is in compliance with guidelines, and that you have complied with the company's (i.e., Twitter's) Terms and Conditions, with appropriate permissions.

Reviewers' comments:

Reviewer's Responses to Questions

**Comments to the Author**

1. Is the manuscript technically sound, and do the data support the conclusions?

Reviewer #1: Partly

Reviewer #2: Partly

Reviewer #3: Yes

Reviewer #4: Yes

2. Has the statistical analysis been performed appropriately and rigorously? 

Reviewer #1: Yes

Reviewer #2: Yes

Reviewer #3: I Don't Know

Reviewer #4: Yes

3. Have the authors made all data underlying the findings in their manuscript fully available?

Reviewer #1: Yes

Reviewer #2: No

Reviewer #3: Yes

Reviewer #4: Yes

4. Is the manuscript presented in an intelligible fashion and written in standard English?

Reviewer #1: Yes

Reviewer #2: Yes

Reviewer #3: Yes

Reviewer #4: Yes

5. Review Comments to the Author

Reviewer #1: Thank you for the opportunity to review “#MaskOn! #MaskOff! Digital polarization of mask-wearing in the United States duringCOVID-19.” This is an interesting study of an obviously important and ongoing subject. At this point, I cannot recommend publication, as I believe the manuscript is too disconnected from some important political phenomena. However, I do hope to see a version of this in print, and I have a number of suggestions below that I hope will help.

My main concern about this manuscript is its disconnection from politics. That is, the authors chart some interesting trends in Twitter use of various mask-related hashtags, but it’s not clear *why* we see the various peaks and valleys in the data, or whether the Twitter patterns are instigating or following other discussions in society. The authors do attempt to correlate these tweet trends with some media headlines, but these prove insufficient. For example, is the dip in hashtag use in early June and then the sudden resurgence a few weeks later related to Covid disease rates? Does it track media coverage of the illness? Is it a function of the George Floyd protests of early June that temporarily dominated the national conversation? More generally, do disease rates or media trends track hashtag usage across the larger time span?

I was also struck by how many more pro-mask tweets there were than anti-mask tweets. As the authors note, this is somewhat consistent with public opinion and public behavior, and Democrats are somewhat more prominent on Twitter than Republicans. However, what is striking to me is how much attention anti-maskers received relative to their numbers. (This might again be an area to bring in a media analysis.) Perhaps this is a function of their controversial nature, which makes for good news stories, or their association with violent activism in state legislatures, or that these messages were largely echoed by the President. But by my read, the most prominent pro-mask tweets outnumbered the most prominent anti-mask tweets by a factor of ten, making it unclear why or how this was even considered a debate. I’m assuming this is an accurate depiction of Twitter activity on masking, although it makes me curious whether anti-maskers use hashtags not detected by the authors or perhaps eschew hashtags altogether. Is it possible to examine tweets in another way looking for content other than hashtags on this topic?

Related to this, I was wondering if anti-maskers had used Twitter to elevate their prominence in excess of their numbers in the population and draw attention to their efforts, or if they were essentially out-gunned on Twitter, relying on other political methods to capture attention.

The authors include some commentary in their discussion section noting the patterns of discussion across mask networks, with mask tweeters mostly tweeting among their own groups, but occasionally engaging those of the opposing viewpoint. The authors suggest, rather fatalistically, that “echo chambers” can induce polarization, but that engagement with people of opposing viewpoints can also induce polarization. Yet surely there are forms of interaction that can depolarize communities. Twitter might not be the best venue for such interaction, but surely some of that occurred. I would encourage the authors to investigate the political communications or political networks literature on this subject.

Related to that topic, I found Figure 4 rather difficult to interpret, and the accompanying text was not terribly Illuminating.

In general, I encourage the authors to situate this research more in the political phenomena surrounding the Twitter usage, rather than just the Twitter usage itself.

I hope these comments are useful.

Reviewer #2: In this paper the authors analyze 412,959 stance-bearing hashtags about mask wearing and identify 35 unique hashtags related to mask wearing. Ultimately, they argue that the use of these hashtags exhibit typical signs of modern partisan polarization within American politics, including conflict that invokes emotional responses and the siloing of ideas into echo chambers. This paper provides a unique approach and a relatively robust look at some interesting data. However, the manuscript is not prepared for publication. The authors lack a clear and articulated theoretical underpinning for their findings that might drive hypothesis. Ultimately, the authors bury their lead at the end and need to engage in significant revision to unpack their claims and the theory they are building upon for this manuscript to be publishable.

I have included a number of notes below. In the interest of illustrating the main issue, that you bury a lead which is actually quite compelling, I preserved all my comments as I wrote them so that you can follow the train of thought a reader might go through.

First, I would like to know more about your qualitative coding scheme for the 35 hashtags, including how many coders participated, was there intercoder reliability across the tests, and finally how did you deal with tags that invoked multiple categories. In addition, was the content of the tweet analyzed in any way to confirm that the hashtags are being used legitimately? In other words, could people sarcastically use the hashtag #maskskill in order to undermine that campaign. I am also interested in your use of the #coverup hashtag. Is that not a mask hashtag? Consider you are writing for a broad audience at PlosOne you really need to unpack this more.

Your discussion of the text content of mask tags is striking, though I am surprised that you include merely the N for the number of tags that, say, employ expletive attributes. I would be much more interested in a clearer discussion of the frequency in use of those attributes relative to other attributes. The existence of these tags is less interesting, their use is far more interesting and speaks volumes to the narrative around mask wearing. The paragraph from lines 162-172 could be divided into two and include this additional discussion of relative frequency of use.

Having arrived at your results section, I am struck by your lack of clear theory and supported hypotheses. I really think there is room before the methodology section to spend more time discussing the current literature on polarization and social media interaction to make your claims more clear.

The paragraph from lines 194-210 is really interesting and compelling with regard to the literature on group-think and echo chambers within social media platforms. This might be a way to incorporate more theoretical grounding within the manuscript. This improved ground could even generate some interesting testable hypothesis regarding the use of tags to 1.) signal identity vs 2.) convince opposition (which implies interaction). Lines 216-228 further emphasize this need for theoretical underpinnings to the argument.

Ultimately, your argument really hits its stride with the first paragraph of the discussion section. Unfortunately, you have buried the lead here. This needs to be up front, with some theory behind it and some expectations in the form of hypotheses. You pack a lot of citations into this paragraph (for example, line 293). These need to be unpacked in a more robust way that produces hypotheses to test.

The paragraph from line 271-283 seems like additional analysis in the discussion section. It seems disjointed placed there. Perhaps it would better fit in the results section as an alternative explanation or as a potential explanation of interaction. What might be interesting to see is if major events, like CDC announcements or COVID increases produced actual conflict as opposed to mere digital echo chambers. This would make better use of this data and add a layer to your analysis.

In terms of your main argument, I think a figure plotting the top 3 pro and anti-mask figures over time would be really interesting. Again, I think you need a theoretical hook here. Perhaps identifying events which drive interaction between the two echo chambers might be an interesting direction. Splitting your Fig 1 and Fig 2 into separate figures makes this less clear.

Ultimately, to reiterate, this paper presents a really interesting analysis without providing any compelling theoretical hook up front to draw the reader in. Further, when you come around to your theoretical claim, you pay passing reference without unpacking the implications of your discoveries on the broader literature on polarization and social media use. This is the most glaring omission in this paper and needs to be corrected before publication.

Other editorial notes:

Table 1: The top line repeats twice.

Equation 4: Why? Most of us know what averages are. This seems excessive.

The resolution on all of your figures is really poor. Be sure this gets resolved if you make it to the proofing phase.

Reviewer #3: The study seems to be well-constructed and carried out. I would not require changes but will point out two possible missed opportunities, which the authors might consider adding at their discretion, depending on how data has been collected and stored. The first is sentiment analysis for the tweets using positive and negative mask hashtags—this seems significant and would add a noteworthy dimension to the data which could not account for other significant demographic vectors (314, 325-30). The second is that the "echo chamber" effect might be further illuminated beyond "interactions between the tweeters" (291) if it were possible to plot, using social network analysis, the follower and mutual followers among and between the hashtag-using communities. I imagine that, if significant enough to be valuable, these might help to describe more fully the behaviors of distinct groups (such as mask makers, trolls, potentially even bot behaviors). Both introduce substantial complexity, but, I believe, potential value added for the study, again, at the authors' discretion.

Reviewer #4: The article addresses an interesting topic regarding the politics of mask wearing during the pandemic. They conduct this analysis by scraping Twitter to find useage of various hashtags that are related to both pro- and anti-mask wearing. The idea and execution of this analysis is well done. I recommend it be published.

6. PLOS authors have the option to publish the peer review history of their article (what does this mean?). If published, this will include your full peer review and any attached files.

Reviewer #1: No

Reviewer #2: No

Reviewer #3: No

Reviewer #4: No

---

## [Author Response · Author response to Decision Letter 0]

25 Mar 2021

Please see attached "Response to reviewer comments".

---

## [Decision Letter · Decision Letter 1]

15 Apr 2021

#MaskOn! #MaskOff! Digital polarization of mask-wearing in the United States during COVID-19

PONE-D-21-03538R1

Dear Dr. Jing-Schmidt,

We’re pleased to inform you that your manuscript has been judged scientifically suitable for publication and will be formally accepted for publication once it meets all outstanding technical requirements.

Kind regards,

Liviu-Adrian Cotfas

Academic Editor

PLOS ONE

Additional Editor Comments (optional):

Reviewers' comments:

Reviewer's Responses to Questions

**Comments to the Author**

1. If the authors have adequately addressed your comments raised in a previous round of review and you feel that this manuscript is now acceptable for publication, you may indicate that here to bypass the “Comments to the Author” section, enter your conflict of interest statement in the “Confidential to Editor” section, and submit your "Accept" recommendation.

Reviewer #1: All comments have been addressed

Reviewer #2: All comments have been addressed

Reviewer #3: All comments have been addressed

Reviewer #4: All comments have been addressed

2. Is the manuscript technically sound, and do the data support the conclusions?

Reviewer #1: (No Response)

Reviewer #2: Yes

Reviewer #3: Yes

Reviewer #4: Yes

3. Has the statistical analysis been performed appropriately and rigorously? 

Reviewer #1: (No Response)

Reviewer #2: Yes

Reviewer #3: Yes

Reviewer #4: Yes

4. Have the authors made all data underlying the findings in their manuscript fully available?

Reviewer #1: (No Response)

Reviewer #2: Yes

Reviewer #3: Yes

Reviewer #4: Yes

5. Is the manuscript presented in an intelligible fashion and written in standard English?

Reviewer #1: (No Response)

Reviewer #2: Yes

Reviewer #3: Yes

Reviewer #4: Yes

6. Review Comments to the Author

Reviewer #1: I appreciate the work the authors have done here to address my concerns and those of the other reviewers. I am comfortable with the manuscript's publication.

Reviewer #2: Excellent job adding some theoretical rigor to your argument. I appreciated the citations up front. Though I can contest some of your assertions about polarization, this will add greatly to the conversation.

Reviewer #3: I saw my comments and other reviewers' were addressed in this revision; my suggestion about social network analysis was said not to be easily implemented, and its value might not have merited the effort.

Reviewer #4: The authors have addressed my concerns and the manuscript should be published. Thank you for the opportunity to review this paper.

7. PLOS authors have the option to publish the peer review history of their article (what does this mean?). If published, this will include your full peer review and any attached files.

Reviewer #1: No

Reviewer #2: No

Reviewer #3: No

Reviewer #4: No

---

## [Editor Report · Acceptance letter]

19 Apr 2021

PONE-D-21-03538R1 

#MaskOn! #MaskOff! Digital polarization of mask-wearing in the United States during COVID-19 

Dear Dr. Jing-Schmidt:

I'm pleased to inform you that your manuscript has been deemed suitable for publication in PLOS ONE. Congratulations! Your manuscript is now with our production department. 

Kind regards, 

on behalf of

Dr. Liviu-Adrian Cotfas 

Academic Editor

PLOS ONE